# Gesture-Controlled Robotic Arm for Agricultural Harvesting Using a Data Glove with Bending Sensor and OptiTrack Systems

**DOI:** 10.3390/mi15070918

**Published:** 2024-07-16

**Authors:** Zeping Yu, Chenghong Lu, Yunhao Zhang, Lei Jing

**Affiliations:** 1Graduate School of Computer Science and Engineering, University of Aizu, Aizuwakamatsu 965-8580, Japan; d8251103@u-aizu.ac.jp (Z.Y.); d8222104@u-aizu.ac.jp (C.L.); m5272029@u-aizu.ac.jp (Y.Z.); 2Department of Computer Science and Engineering, University of Aizu, Aizuwakamatsu 965-8580, Japan

**Keywords:** gesture control, robotic arm, agricultural harvesting, data glove, bending sensors, OptiTrack, machine learning, CNN+BiLSTM, spatial tracking, ergonomic design

## Abstract

This paper presents a gesture-controlled robotic arm system designed for agricultural harvesting, utilizing a data glove equipped with bending sensors and OptiTrack systems. The system aims to address the challenges of labor-intensive fruit harvesting by providing a user-friendly and efficient solution. The data glove captures hand gestures and movements using bending sensors and reflective markers, while the OptiTrack system ensures high-precision spatial tracking. Machine learning algorithms, specifically a CNN+BiLSTM model, are employed to accurately recognize hand gestures and control the robotic arm. Experimental results demonstrate the system’s high precision in replicating hand movements, with a Euclidean Distance of 0.0131 m and a Root Mean Square Error (RMSE) of 0.0095 m, in addition to robust gesture recognition accuracy, with an overall accuracy of 96.43%. This hybrid approach combines the adaptability and speed of semi-automated systems with the precision and usability of fully automated systems, offering a promising solution for sustainable and labor-efficient agricultural practices.

## 1. Introduction

With the continuous growth of the global population, food issues around the world are becoming increasingly severe. The United Nations Department of Economic and Social Affairs reported that the world population reached 7.942 billion in 2022, with projections indicating increases to 8.512 billion by 2030 and 9.687 billion by 2050 [1]. Concurrently, data from the International Labor Organization reveal a downward trend in the employment in agriculture, forestry, and fishing, decreasing from approximately 968.475 million in 2010 to about 855.386 million by 2020 [2]. This decline is further compounded by demographic shifts in developed nations, particularly the aging population issue, which is steadily eroding the agricultural workforce. A case in point is Japan, where, based on Agricultural Census data, the number of individuals engaged in agriculture dropped from 1.757 million in 2015 to 1.363 million in 2020, and the percentage of those aged 60 and above rose from 78.7% to 79.9% [3]. The confluence of a shrinking and aging agricultural workforce, against the backdrop of a burgeoning global population, portends a looming food crisis.

Mechanization and automation have emerged as pivotal solutions to the crisis faced in agriculture. Statistical analysis reveals a significant reduction in labor requirements for China’s three principal cereal crops, decreasing from 13.80 labor days per mu (a unit of area measurement commonly used in China equivalent to approximately 666.67 m^2^) in 1998 to 4.81 labor days per mμ in 2018, marking a substantial decline of 65.14%. In contrast, advancements in the mechanization and automation of the fruit sector lag behind those of the grain industry. Labor inputs per mu for apple production saw a decrease from 48.70 labor days in 1998 to 33.85 labor days in 2018 (a reduction of 14.85 labor days), translating to a modest decline of 30.49%. The Ministry of Agriculture and Rural Affairs of China estimates that the comprehensive mechanization rate for major cereal crops surpassed 80% in 2018 [4].

At this moment, fruit production is still significantly dependent on human labor, despite the potential for mechanization in specific segments of the process. The stages that demand the highest amount of labor include pruning, pollination, bagging, and harvesting, with the latter being the most labor-intensive. The necessity for such extensive manual labor stems, mainly due to the irregular spatial growth patterns of many fruits, hinders the widespread adoption and effectiveness of mechanization and automation technologies. Although numerous studies have been aimed at developing automated systems for fruit production, the operational speed of these systems significantly lags behind human performance, and they often come with a steep learning curve [5]. Therefore, this research focuses on the development of a user-friendly robotic arm controller that enables efficient fruit harvesting, among other agricultural tasks, through gesture recognition for remote control, effectively mimicking human actions.

To achieve the aforementioned objectives, efforts must be concentrated on three fronts. First, we must define and recognize easily executable human hand gestures to control certain behaviors of the robotic arm, such as power on/off or opening/closing of the end effector. Second, it is imperative to accurately capture the movement distance of the human hand and replicate its coordinates for the robotic arm. Third, a system must be established that processes the aforementioned data and sends control commands to the robotic arm.

Initially, to identify human hand gestures, we fabricated a data glove equipped with bending sensors at the joints of the hand. This glove is capable of measuring the voltage changes in the bending sensor detection circuit and transmitting the data to a server. Subsequently, by placing three reflective markers on the glove, we employed an OptiTrack system to acquire motion data. The associated Motive (Body 3.0.1 Final) software computes the coordinate data and dispatches them to the server. Finally, by executing custom-developed data transmission and processing software on the server, machine learning and deep learning algorithms are utilized to recognize the bending sensor data transmitted by the data glove. This recognition process identifies hand gestures and sends the corresponding control commands to the robotic arm. Concurrently, the coordinate data received from the OptiTrack system are converted and relayed as coordinate control commands to the robotic arm.

Existing solutions and their limitations can be summarized as follows:Fully automated systems: These systems promise reduced human labor but suffer from slow operational speeds, inefficiencies in adapting to various crop types and environmental conditions, and high initial and maintenance costs.IMU-based data gloves: While enhancing interactivity between human operators and robotic systems, these gloves are prone to drift errors and require complex calibration, affecting precision crucial for tasks like fruit harvesting.OptiTrack systems: Despite offering superior accuracy and lower latency, they face challenges in environments with potential obstructions to the line of sight, leading to inaccuracies in gesture recognition.

The contributions in this paper can be summarized as follows:An integrated system combining bending sensors and an OptiTrack system was developed for precise gesture recognition and spatial tracking.The convenience and accuracy of robotic control were enhanced through advanced hand gesture recognition.By leveraging the complementary strengths of bending sensors and the OptiTrack system, issues associated with IMU-based data gloves, such as spatial coordinate drift, were mitigated, and the keypoint loss problem in OptiTrack’s hand movement tracking was addressed.

The remainder of this paper is structured as follows. This work is compared with other data gloves and robotic arm control systems, discussing the systematic design choices made in both hardware and software aspects. Moreover, the paper evaluates the gesture accuracy of several research subjects and the response speed of the robotic arm to movement and gesture recognition, followed by a discussion of the results. The final chapter summarizes the main findings in this work and provides an outlook on future research.

## 2. Related Works

The automation of fruit harvesting has seen various technological interventions, primarily categorized into fully automated systems and semi-automated systems that incorporate human operators. The development of fully automated fruit harvesting robots, such as those discussed by Yoshida et al. (2022) [5] and Majeed et al. (2022) [6], has primarily focused on providing a complete mechanization solution that promises to reduce human labor. However, these systems are often hampered by slow operational speeds and inefficiencies, particularly when adapting to diverse types of crops or varying environmental conditions. The high initial setup and maintenance costs, coupled with their limited flexibility, render these systems less feasible for widespread adoption.

In contrast, data gloves equipped with Inertial Measurement Units (IMUs) represent a significant advancement in enhancing the interactivity between human operators and robotic systems. Lu et al. (2023) [7] provided detailed hand-tracking capabilities such devices, which offer real-time spatial tracking and have been utilized in various applications, ranging from virtual reality to interactive robotics. Despite their versatility, IMUs are prone to drift errors and often require complex calibration procedures to maintain accuracy, as highlighted by Lin et al. (2018) [8] and Rodić et al. (2023) [9]. These limitations significantly impact the precision required for tasks like fruit harvesting, where delicate handling and exact positioning are crucial.

To address the spatial tracking issues inherent in IMU systems, OptiTrack systems have been employed due to their use of high-precision cameras and reflective markers for motion tracking. This technology offers superior accuracy and lower latency compared to IMU-based systems, making it suitable for applications requiring high precision. However, as noted by the Comparative Analysis of OptiTrack Motion Capture Systems (2018) [10], these systems can face challenges in environments where the line of sight can be obstructed, leading to inaccuracies in gesture recognition. Such occlusions, common in outdoor agricultural settings, can significantly reduce the efficacy of tracking systems.

Table 1 below provides a comparison of various data gloves based on their performance metrics for hand gesture recognition. It includes details about the type of sensors used for detecting hand gestures and positioning, the cost of the gloves, and the corresponding references. While commercial systems such as HaptX, Manus Meta, and TactGlove DK2 may appear more robust, they are not necessarily smaller or cheaper. Our glove costs approximately USD 100 and could be even more affordable in mass production. Additionally, it is important to note that TactGlove DK2, while being the least expensive option, does not inherently possess any gesture recognition or positioning capabilities. It relies entirely on external VR cameras for these functions, which contributes to its lower price.

In recent years, deep learning algorithms have made significant breakthroughs in sensor-based gesture recognition. For instance, Guan Yuan et al. [14] proposed a hand gesture recognition system using a deep feature fusion network based on wearable sensors. This glove includes two armbands and an integrated three-dimensional bending sensor capable of capturing fine-grained movements of the entire arm and all finger joints, with an LSTM model using fused feature vectors as input, yielding excellent results. Yongfeng et al. [15] proposed a dynamic gesture recognition algorithm (DGDL-GR), which achieved promising results by capturing finger movements and bending data. Jiawei Wu et al. [15] further advanced this field by introducing a gesture recognition method that combines Convolutional Neural Networks (CNNs) and Bidirectional Long Short-Term Memory networks (BiLSTMs), incorporating an attention mechanism to enhance recognition accuracy and robustness. Additionally, Yang Song et al. [16] utilized a wearable wrist sensor made from flexible pressure sensors integrating CNN and BiLSTM models for gesture recognition, demonstrating the potential of this approach. These methods not only capture complex finger movements and bending degrees but also achieve precise gesture recognition through deep learning algorithms. The combination of the powerful feature extraction capabilities of CNNs and the sensitivity of BiLSTMs to temporal information significantly enhances the performance of gesture recognition systems based on bending sensors. These advancements highlight the crucial role of deep learning algorithms in improving the accuracy and robustness of gesture recognition systems.

Furthermore, recent advancements in machine learning and sensor technology have opened new avenues for enhancing these hybrid systems. As demonstrated by Ran Bi (2023) [17], integrating machine learning algorithms with sensor data can significantly improve the adaptability and efficiency of gesture recognition systems, paving the way for more responsive and intuitive control mechanisms in agricultural robotics.This hybrid approach addresses the critical shortcomings of fully automated systems, such as adaptability and speed, while overcoming the spatial accuracy and occlusion issues prevalent in traditional data glove systems. For example, combining vision and bending sensor data can enhance recognition performance. The multi-modal fusion gesture recognition system proposed by Lu et al. (2021) [18] successfully integrates camera data and data glove data, improving the recognition rate of gestures under occlusion. However, the introduction of video data results in higher computational costs. The integration of these technologies presents a promising avenue for developing more efficient and flexible robotic solutions for fruit harvesting, potentially transforming agricultural practices to be more sustainable and less labor-intensive.

Moreover, the human–robot interaction (HRI) aspect of semi-automated systems, which is crucial for tasks requiring high levels of precision and adaptability, has not been fully explored. Current systems often do not account for the ergonomic and cognitive loads placed on human operators, which can affect the overall efficiency and adoption of these technologies. As highlighted by Rodić et al. (2023) [9], enhancing the intuitive aspects of human–machine interfaces and reducing the cognitive burden through better design and integration of feedback mechanisms are critical areas needing attention.

In light of these challenges, this study proposes a novel hybrid approach that combines the OptiTrack system with bending sensors integrated into a data glove. This method not only leverages the high spatial accuracy of OptiTrack but also incorporates the flexibility and resilience of bending sensors to provide robust gesture recognition. even in complex and dynamic agricultural environments. This approach effectively bridges the gap between the adaptability and speed of fully automated systems and the accuracy and usability of semi-automated systems.

## 3. System Architecture and Design

### 3.1. Overview

Figure 1 shows the system architecture, which consists of the following three main components: the robotic arm controller, the server, and the robotic arm. The robotic arm controller consists of the data glove, six OptiTrack cameras, and a video monitor. The data glove has ten bending sensors for gesture recognition and three reflective markers for captured hand coordinates. The server side consists of three different programs; Motive is responsible for processing the video data from the OptiTrack camera and converting it into coordinate data, the gesture recognition program is responsible for decoding the received data and sending the recognized results to the data processing program, the data processing program sends the control commands to the robotic arm based on the recognized results, and the data between the programs are exchanged through sockets. The arm is controlled by a built-in Raspberry Pi, which operates six joint motors and end effectors after receiving the control commands. The robotic arm is also equipped with a Wi-Fi camera, which is used to remotely transmit the real-time image to the video monitor of the controller.

### 3.2. Hardware Design

#### 3.2.1. Data Glove

The data glove contains ten BS-65 bending sensors from Sensia Technology, three reflective markers provided by the OptiTrack system, and a custom-built board with resistance detection and charging/discharging circuits powered by an ESP32-S3 micro-controller (Espressif Systems, Shanghai, China).

The response curve of the BS-65 bending sensor is shown in Figure 2. Its resistance is 20 kΩ ± 10% when unbent and changes from about +250% to −40% when bent. We followed the method described by A. K. Bose et al. [19] to test the impact of stress on this bending sensor. The calculated results are presented as follows: at 5 N, ΔR/R=0.585; at 10 N, ΔR/R=1.269; at 15 N, ΔR/R=2.253; at 20 N, ΔR/R=3.194. To avoid the influence of stress on the bending sensor’s readings, we fixed only one end of the sensors to the glove during its fabrication, allowing the sensor to move freely within a certain range.

Figure 3 shows the custom-designed ESP32 board, which includes battery charging/discharging management, Wi-Fi data transmission, Real-Time Clock (RTC), and Analog-to-Digital Conversion (ADC) functions. The board’s through holes are specially designed. It has larger, oval-shaped solder pads and holes to facilitate easy attachment to the data glove through sewing.

Figure 4 shows the data glove equipped with bending sensors and OptiTrack reflective markers. The reflective markers, small spheres made of reflective material, efficiently reflect infrared light emitted by the OptiTrack system’s cameras, allowing for precise position tracking. These markers are attached to the data glove using Velcro straps. The glove itself is integrated with bending sensors sewn into the fabric to accurately measure hand movements.

Figure 5 shows the resistance measurement circuit on the ESP32 board; Figure 5a is an inverted signal amplifier circuit used to convert the change in measured resistance to a change in voltage for easy measurement with the following formula:(1)$$V_{out}=-\frac{R_{fb}}{R_{bs}}\times V_{ref}$$
where $V_{out}$ represents the output voltage, which is directly connected to the AD conversion pin of the ESP32; $R_{bs}$ is the resistance of the bending sensor; $R_{fb}$ is the feedback resistor of the inverting signal amplification circuit; and $V_{ref}$ is the reference voltage at the output of the circuit shown in Figure 5b.

The reference voltage circuit consists of a resistor divider circuit and a voltage follower, and the value of $V_{ref}$ is calculated by the following formula:(2)$$V_{ref}=\frac{R_{1}}{R_{1}+R_{2}}\times V_{dd}$$
where $V_{dd}$ is 3.3 V, serving as the system’s supply voltage. The system is powered by a battery and includes an LDO chip (Texas Instruments TLV75733). $R_{1}$ and $R_{2}$ are the resistors in the voltage divider circuit, with resistance values of 1.8 kΩ and 200 Ω, respectively.

The circuit depicted in Figure 5 was simulated using PSpice for TI 17.4-2023, with the results presented in Figure 6. The simulations varied the feedback resistor ($R_{fb}$) at values of 90 kΩ, 110 kΩ, and 130 kΩ, while the bending sensor resistance ($R_{bs}$) ranged from 10 kΩ to 60 kΩ. The output voltage ($V_{out}$) exhibited significant differences under these conditions. Given that the minimum resistance of $R_{bs}$ is approximately 14 kΩ, utilizing a 130 kΩ $R_{fb}$ might position the operating point within a cutoff region, while a 90 kΩ $R_{fb}$ could result in insufficient sensitivity when $R_{fb}$ exceeds 50 kΩ. After a comprehensive evaluation, an $R_{fb}$ value of 110 kΩ was determined to optimize both the measurement range and sensitivity, ensuring a balanced operational profile.

#### 3.2.2. Robotic Arm

The utilized robotic arm is a commercially available model from Elephant Robotics. As illustrated in Figure 7, myCobot 320 Pi adopts a Raspberry Pi microprocessor. Its body weights 3 kg with a load of 1 kg and a working radius of 320 mm. A total of six degrees of freedom can be achieved using this robotic arm. The end effector shown in Figure 10 would be considered an additional degree of freedom, as it can close its claws there by cutting in the desired way.

### 3.3. Software Design

As illustrated in Figure 8, due to variations in sampling rates across different devices, linear processing methods can lead to significant latency. To reduce the overall system delay, the software design is segmented into four specialized programs.

The ESP32 board program (C#, Visual Studio 2022 IDE) leverages the ADC functionality of the ESP32 and the circuit detailed in Figure 5 to read voltage values from bending sensors and transmit them to the server. The deep learning program(Python 3.12.4) is tasked with importing real-time data into a pre-trained model and exporting the outcomes. The data process program(C, Arduino IDE 2.3.2) oversees data processing and exchange within the entire system, transmitting sensor data from the data glove to the deep learning program and converting the results from the deep learning program into control commands for the robotic arm program. Additionally, it translates coordinate data from OptiTrack into further control commands. The robotic arm program forwards these commands to the motor and end effector, calculating delays via a connected RTC. All four components are designed to operate simultaneously upon system startup.

### 3.4. Proportion Calculation for Elimination of Systematic Errors

To address and eliminate systematic errors in the robotic arm, it is essential to account for any potential delay between the coordinates of the human hand and the robotic arm. This delay can be calculated based on the sampling frequency and time differences observed in the data.

First, the data are aligned by shifting the robotic arm’s coordinate data forward by the calculated number of frames. This temporal alignment ensures that the data from the robotic arm correspond accurately to the data from the human hand.

To further eliminate systematic errors, the proportion factors between the coordinates of the robotic arm and the human hand are computed using linear regression. This method involves finding the best-fitting line that minimizes the sum of squared differences between the observed values and the values predicted by the model. The linear regression equations are
(3)$$\begin{array}{cc} x_{m} & =k_{x}\cdot x_{h}+b_{x}\\ y_{m} & =k_{y}\cdot y_{h}+b_{y}\\ z_{m} & =k_{z}\cdot z_{h}+b_{z}, \end{array}$$
where $x_{m}$, $y_{m}$, and $z_{m}$ are the coordinates of the robotic arm; $x_{h}$, $y_{h}$, and $z_{h}$ are the coordinates of the human hand; $k_{x}$, $k_{y}$, and $k_{z}$ are the proportion factors; and $b_{x}$, $b_{y}$, and $b_{z}$ are the biases for the X, Y, and Z coordinates, respectively.

The linear regression model is fitted to determine the proportion factors and biases. This fitting process helps to precisely model and correct the relationship between the coordinates of the robotic arm and the human hand, thereby eliminating systematic errors in the robotic arm’s movements.

### 3.5. Deep Learning Models for Gesture Recognition

In the development of the hand gesture recognition system for robotic control, experiments were conducted with three deep learning structures, namely LSTM, BiLSTM, and a combination of CNN with BiLSTM.

LSTMs are ideal for this application because they handle sequences, such as the readings from the glove’s bend sensors, by retaining information for extended durations. This capability is crucial for predicting sequences of hand positions.

BiLSTMs build on LSTMs by analyzing sequences both from the beginning to the end and vice-versa. This two-way analysis is better for recognizing complex gestures, as it considers what comes before and after in a sequence.

This combined model, CNN+BiLSTM, starts with CNN layers that pull out important spatial features from sensor data, then passes these on to BiLSTM layers. This mix is particularly effective at capturing patterns related to both space and time, which helps make recognition more accurate and reliable.

As shown in Figure 9, the sensor data first pass through a 1D CNN layer, which detects important spatial patterns, specifically the relationship between the curved sensors in a frame. Batch normalization ensures that the model learns efficiently and consistently.

Next comes a Leaky ReLU activation function, which introduces non-linearity. This means the network can learn more complex patterns, which is essential for differentiating subtle hand movements.

Two BiLSTM layers form the core of the setup. They are adept at understanding long-term patterns in the sensor data, considering what comes both before and after in a sequence. This provides a strong foundation for accurately classifying gestures.

Another batch normalization and Leaky ReLU set up the final BiLSTM layer. This is crucial when recognizing complex gestures that change significantly over time. This hybrid approach effectively captures both spatial and temporal features, significantly improving the recognition accuracy and robustness against noisy data. However, due to the increase in model expressiveness, the risk of overfitting is exacerbated.

## 4. Experiments and Results

The system primarily achieves the following two functions: following the trajectory of hand movements with the end effector of the robotic arm and recognizing hand gestures to execute corresponding functions. Therefore, the validation of the two experiments focuses on the spatial trajectory error between the hand and the robotic arm and the accuracy of hand gesture recognition.

### 4.1. Experiment 1: Spatial Trajectory Error

In this experiment, data were collected for one minute of back-and-forth movement along each motion axis, as shown in Figure 10, totaling 3 min and including 1800 frames of data. For the spatial trajectory, the displacement ratio between the hand and the robotic arm was adjusted to be one to one. A marker was also placed on the end-effector of the robotic arm to track its trajectory. The errors between the trajectories of the hand and the robotic arm were then compared.

To address and eliminate systematic errors in the robotic arm, as shown in Figure 11, it is essential to evaluate the stability of the delay between the coordinates of the human hand and the robotic arm. Based on the collected delay data, several statistical measures were computed to assess this stability. The mean delay was found to be 0.325 s, representing the average time difference between the movements of the human hand and the corresponding response of the robotic arm. The standard deviation of the delays was 0.0625 s, indicating the amount of variation or dispersion from the average delay. A lower standard deviation suggests that the delays are more consistent. Additionally, the interquartile range (IQR) of the delays was calculated to be 0.09 s, reflecting the range within which the middle 50% of the delay values fall. The IQR is a robust measure against outliers, providing a clear picture of the variability in the delays. The first quartile (Q1) was 0.275 s, and the third quartile (Q3) was 0.365 s. These statistical measures collectively demonstrate the effectiveness of the delay correction process in achieving stable and predictable synchronization between the human hand and the robotic arm.

To evaluate the spatial trajectory error between the hand and the mechanical hand, the following three metrics were calculated: mean Euclidean distance, the standard deviation of the Euclidean distance, and root mean square error (RMSE). These metrics provide a quantitative assessment of the accuracy and precision of the mechanical hand’s movements in replicating the intended hand gestures.

The Euclidean distance measures the straight-line distance between corresponding points on trajectories of the hand and the mechanical hand. It is a direct measure of the deviation at each time point. The Euclidean distance is calculated using the following formula:(4)$$E=\sqrt{{\left(x_{1}-x_{2}\right)}^{2}+{\left(y_{1}-y_{2}\right)}^{2}+{\left(z_{1}-z_{2}\right)}^{2}}$$

RMSE provides an aggregated measure of the overall error by considering the squared differences between the trajectories of the hand and mechanical hand, thereby giving more weight to larger errors. RMSE is calculated using the following formula:(5)$$\mathrm{RMSE}=\sqrt{\frac{1}{n}\sum_{i=1}^{n}\left[{\left(x_{1_{i}}-x_{2_{i}}\right)}^{2}+{\left(y_{1_{i}}-y_{2_{i}}\right)}^{2}+{\left(z_{1_{i}}-z_{2_{i}}\right)}^{2}\right]}$$
where $\left(x_{1},y_{1},z_{1}\right)$ are the coordinates of the hand and $\left(x_{2},y_{2},z_{2}\right)$ are the coordinates of the end effector.

The results of this experiment demonstrate that the mean Euclidean distance between the hand and the mechanical hand is 0.0131 m, while the root mean square error (RMSE) is 0.0095 m. Additionally, the standard deviation of the Euclidean distance is 0.0100 m.

These results indicate a high level of precision in the mechanical hand’s ability to replicate the intended hand movements, with minimal deviation and error.

### 4.2. Experiment 2: Hand Gesture Recognition Accuracy

In the second experiment, the accuracy of hand gesture recognition was evaluated. A dataset was amassed comprising seven types of hand gestures with a temporal dimension, as shown in Figure 12. These gestures include rest, show 1 (index finger up), show 2 (index and middle fingers up), claw, fist, pinch with index finger and thumb, and all-finger pinch. Each gesture was recorded at a sampling rate of 100 Hz for a 2 s duration, ensuring a variety of temporal states and initiating positions were captured. Each gesture was performed ten times under ten distinct conditions, resulting in 100 data samples per gesture and a comprehensive total of 1400 s of data.

The collected data were meticulously divided into training, validation, and testing sets with a ratio of 6:2:2. This split was designed to provide a robust training framework while retaining sufficient data for effective model validation and testing.

In terms of model training, a low learning rate of 0.00001 was set to fine tune the network’s adjustments during learning. ’Sparse categorical cross entropy’ was utilized as the loss function due to its suitability for multi-class classification tasks. The Adam optimizer facilitated the learning process over 300 epochs, with dropout implemented to combat overfitting. The model learned to discern the subtleties between different hand gestures, which is critical for accurate classification.

This meticulous approach to data collection and the deliberate choice of model parameters were fundamental in developing a hand gesture recognition system capable of interpreting nuanced human gestures for the control of robotic hands.

In this experimental setup, each model was trained and validated on a dataset comprising seven distinct hand gestures. The CNN+BiLSTM model outperformed the standalone LSTM and BiLSTM models in terms of accuracy and processing speed. This superior performance can be attributed to its ability to leverage both spatial and temporal dynamics, which is critical for dynamic and accurate gesture recognition in real-time applications.

The performance of three different models for hand gesture recognition was compared, namely CNN+BiLSTM, BiLSTM, and LSTM. The CNN+BiLSTM model exhibited the highest performance, achieving an accuracy of 0.9500, a precision of 0.9531, a recall of 0.9500, and an F1 score of 0.9503. As shown in Table 2, this model, with 365,511 total parameters, effectively recognized and classified the various hand gestures. The combination of convolutional neural networks (CNNs) for feature extraction and bidirectional long short-term memory (BiLSTM) networks for temporal dependencies allowed the model to capture the intricate details of hand gestures, thus demonstrating its superior capability in this task.

The BiLSTM model was less complex than the CNN+BiLSTM model yet still maintained high classification performance. The reduction in model complexity did not significantly compromise its accuracy, making it a viable option for applications requiring a balance between performance and computational efficiency.

In contrast, the LSTM model, having only 19,655 parameters, demonstrated the limitations of a simpler architecture in accurately recognizing hand gestures. While it required less computational power, its significantly lower performance highlights the necessity of more sophisticated models like CNN+BiLSTM or BiLSTM for tasks demanding high precision and reliability in gesture recognition.

The confusion matrix shows that the CNN+BiLSTM model achieves high accuracy across most hand gesture categories, with gestures 0, 4, 5, and 6 classified correctly 100% of the time. Minor misclassifications occur in gestures 1, 2, and 3, indicating similarities that make them harder to distinguish. Overall, the model demonstrates robust performance, with a majority of gestures being accurately classified.

As shown in Figure 13 and Figure 14, the confusion matrix shows that the CNN+BiLSTM model achieves high accuracy across most hand gesture categories, with gestures 0, 4, 5, and 6 classified correctly 100% of the time. Minor misclassifications occur for gestures 1, 2, and 3, indicating similarities that make them harder to distinguish. Overall, the model demonstrates robust performance, with a majority of gestures being accurately classified.

The training and validation loss and accuracy graphs indicate effective learning with minimal overfitting. Both training and validation loss decrease smoothly, while the accuracy curves rise and stabilize close to each other, reflecting strong generalization to unseen data. These results highlight the model’s capability for accurate and reliable hand gesture recognition, making it suitable for real-world applications.

## 5. Discussion and Conclusions

The results of this study indicate that combining bending sensors and OptiTrack systems in a data glove to control a robotic arm in agricultural harvesting offers significant advantages over traditional methods. This hybrid approach addresses the critical shortcomings of fully automated systems, such as adaptability and speed, while overcoming the spatial accuracy and occlusion issues prevalent in traditional data glove systems. The integration of these technologies presents a promising avenue for developing more efficient and flexible robotic solutions for fruit harvesting, potentially transforming agricultural practices to be more sustainable and less labor-intensive.

Experiment 1 demonstrated that the spatial trajectory error between the hand and the robotic arm was minimal. By evaluating the Euclidean distance and root mean square error (RMSE), it was found that the robotic arm was able to closely follow the hand movements with a mean Euclidean distance of 0.0131 m, a standard deviation of the Euclidean distance of 0.0100 m, and an RMSE of 0.0095 m. This high level of precision indicates that this system can accurately replicate intended hand movements, which is crucial for delicate tasks like fruit harvesting.

The CNN+BiLSTM model outperformed both the standalone LSTM and BiLSTM models, achieving the highest accuracy, precision, recall, and F1-score in Experiment 2. This superior performance can be attributed to the model’s ability to leverage both spatial and temporal dynamics, which is crucial for dynamic and accurate gesture recognition in real-time applications. Confusion matrix analysis revealed high accuracy across most hand gesture categories, with minor misclassifications occurring in gestures that are inherently similar, indicating the robustness of the model.

One of the key insights from this study is the necessity to balance model complexity and computational efficiency. While the CNN+BiLSTM model demonstrated the highest performance, the BiLSTM model, despite being less complex, maintained high classification performance, making it a viable option for applications where computational resources are limited. In contrast, the LSTM model, with its simpler architecture, was less effective in accurately recognizing hand gestures, highlighting the need for more sophisticated models in tasks demanding high precision and reliability.

The ergonomic aspect of the data glove was also a critical consideration. By integrating OptiTrack systems with bending sensors in a user-friendly design, the goal was to minimize the cognitive load on operators while maximizing the system’s adaptability and efficiency. This integration not only improved gesture recognition accuracy in complex environments but also enhanced the ergonomic experience, making it more practical for everyday use by agricultural workers.

In conclusion, this research demonstrates that a gesture-controlled robotic arm using a data glove with bending sensors and OptiTrack systems is a feasible and effective solution for agricultural harvesting. The hybrid approach leverages the strengths of both technologies, offering high precision, adaptability, and user-friendly operation. The CNN+BiLSTM model proved to be the most effective in gesture recognition, underscoring the importance of combining spatial and temporal analysis for accurate and reliable performance.

Future research should focus on further refining the ergonomic design of the data glove to reduce operator fatigue and enhance usability. Additionally, exploring other machine learning models and integrating advanced sensor technologies could further improve the system’s accuracy and efficiency. Future work should aim to integrate OptiTrack with IMUs, enabling the system to switch to IMU-based operation when visual localization is lost, ensuring continuous functionality. In addition to this integration, the system might face challenges such as occlusion and varying light conditions, affecting the reflective markers within the indoor environment. For instance, equipment or other objects might block the line of sight to the markers, or shadows cast by moving equipment could impact marker visibility. To overcome these challenges, employing multi-sensor fusion technology, such as by combining data from IMUs and cameras, could enhance the system’s robustness. Furthermore, integrating temperature and humidity sensors to provide real-time calibration of the MEMS bending sensors can help eliminate the influence of environmental factors on sensor accuracy, ensuring consistent performance. Finally, field trials in diverse agricultural environments are essential to validate the system’s performance under real-world conditions and identify any areas for improvement.

By addressing these aspects, more sophisticated, reliable, and user-friendly robotic solutions can be developed that will significantly contribute to the automation of agricultural tasks, thereby addressing labor shortages and increasing the efficiency and sustainability of food production.

## Figures and Tables

**Figure 1 micromachines-15-00918-f001:**
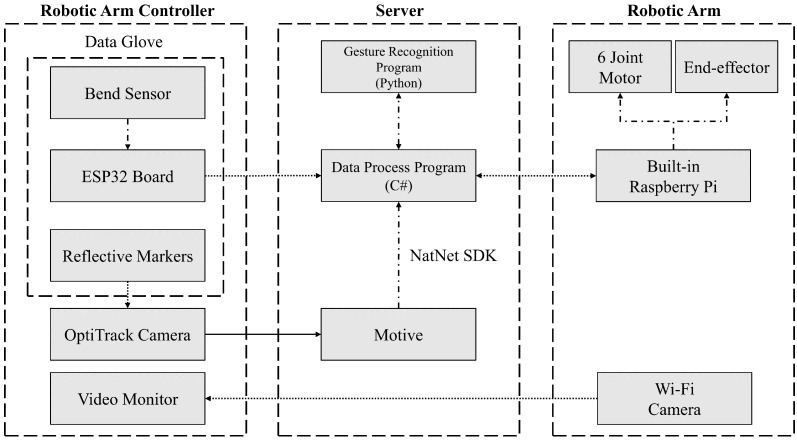
System block diagram. Solid arrows means that the data are transmitted through a wire, dashed arrows mean that the data are transmitted wirelessly, and dotted arrows mean that the data are transmitted within the system.

**Figure 2 micromachines-15-00918-f002:**
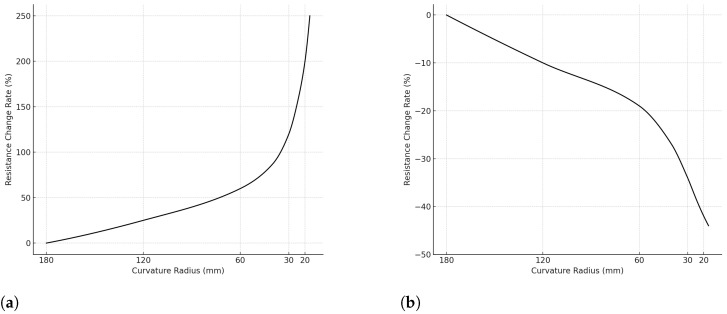
BS-65 bending sensor response curve. When bent in the direction that stretches the sensor surface, the resistance increases. When bent in the direction that compresses the sensor surface, the resistance decreases. (**a**) Tensile strength of the sensor surface. (**b**) Compressive strength of the sensor surface.

**Figure 3 micromachines-15-00918-f003:**
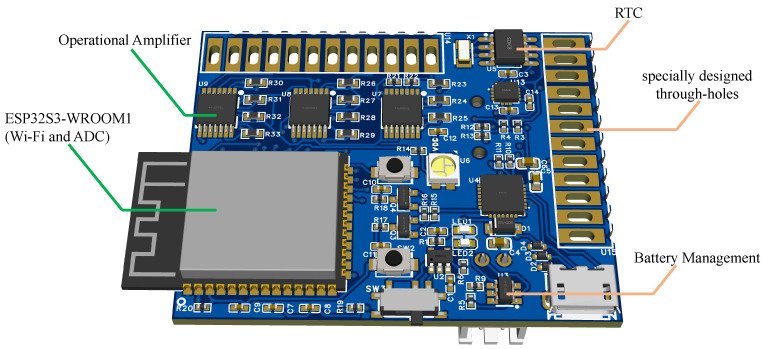
The custom-designed ESP32 board. The board features an ESP32S3-WROOM1 module (left) for Wi-Fi and ADC functionalities, operational amplifiers (U7, U8, and U9), a real-time clock (U5), and a battery management system (U3).

**Figure 4 micromachines-15-00918-f004:**
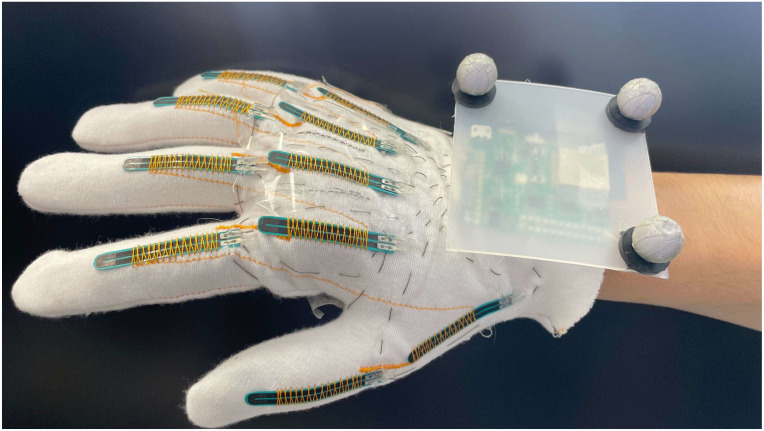
Bending sensor data glove, consisting of ten bend sensors fixed on cloth.

**Figure 5 micromachines-15-00918-f005:**
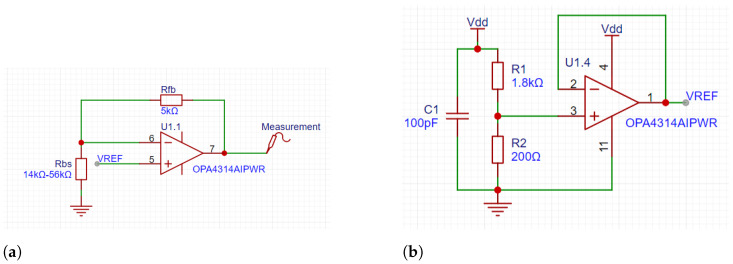
The resistance measurement circuit; these circuits are used to measure and convert resistance changes in the bending sensor to voltage changes for easy measurement. (**a**) The inverting signal amplifier circuit. (**b**) The reference voltage circuit.

**Figure 6 micromachines-15-00918-f006:**
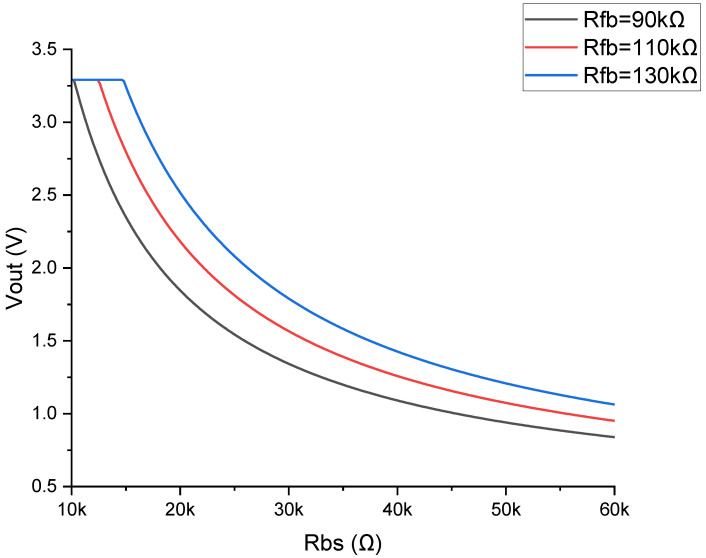
The output voltage ($V_{out}$) versus the resistance of the bending sensor ($R_{bs}$) for different feedback resistor values ($R_{fb}$) in the inverting signal amplifier circuit. The graph shows the response curves for $R_{fb}$ values of 90 kΩ (black), 110 kΩ (red), and 130 kΩ (blue).

**Figure 7 micromachines-15-00918-f007:**
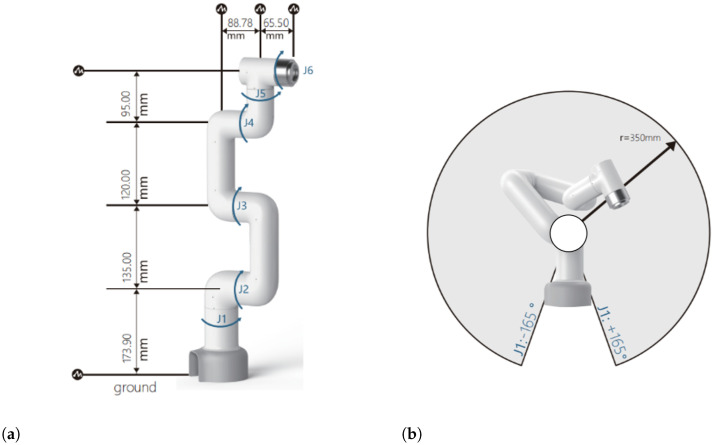
Schematic diagram of the robotic arm showing its dimensions and range of motion. (**a**) The figure illustrates the lengths of each segment of the robotic arm and their respective joints (J1 to J6), with measurements provided in millimeters. (**b**) The figure illustrates the range of motion.

**Figure 8 micromachines-15-00918-f008:**
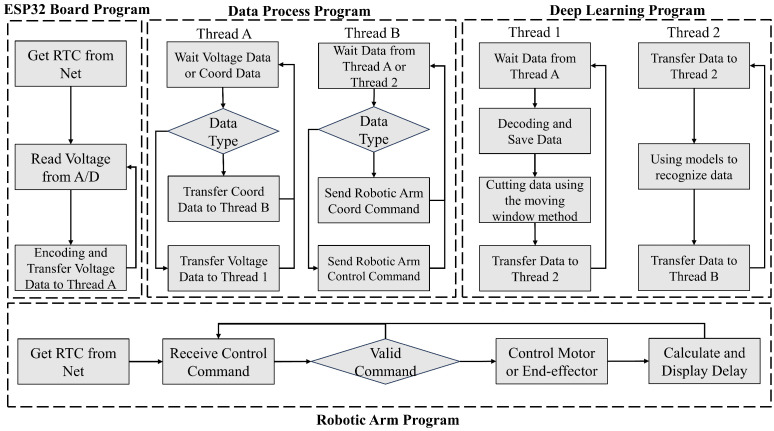
System architecture and data flow for the gesture-controlled robotic arm. The architecture comprises the following four main programs: the ESP32 board program, data processing program, deep learning program, and robotic arm program.

**Figure 9 micromachines-15-00918-f009:**
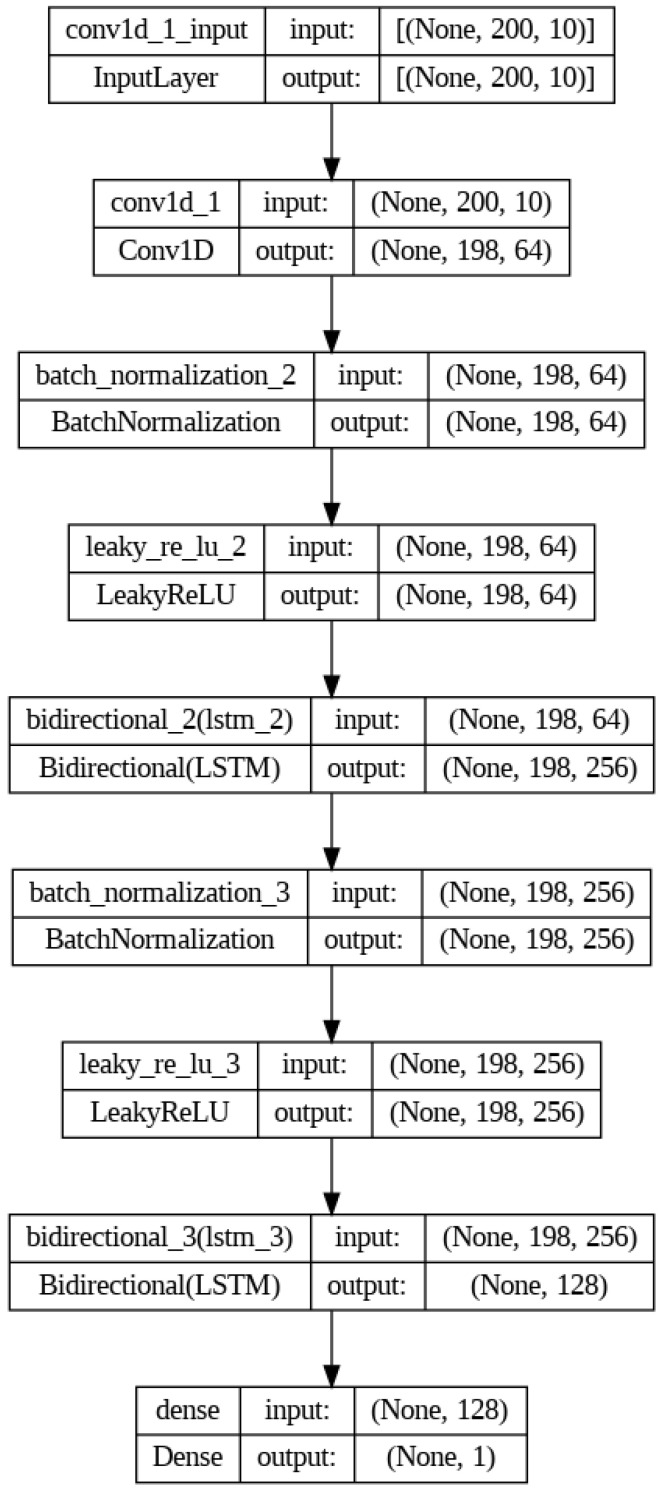
CNN+BiLSTM network architecture.

**Figure 10 micromachines-15-00918-f010:**
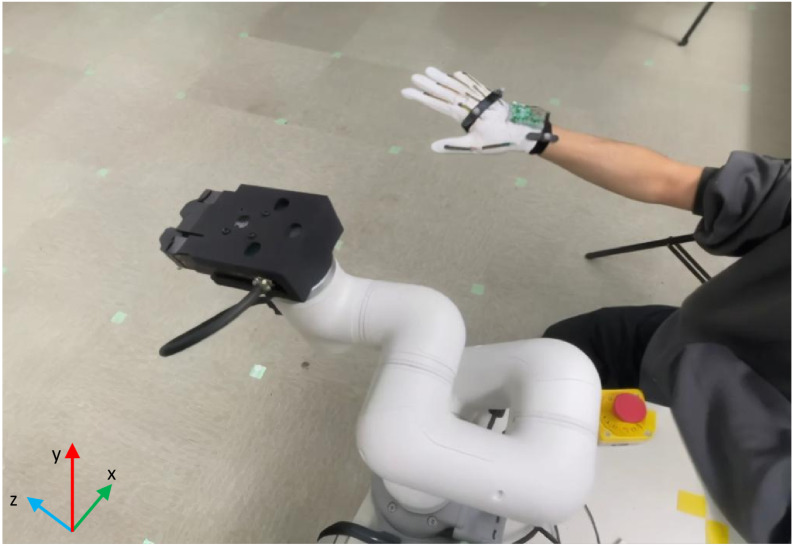
Experimental setup showing a hand wearing the data glove and the robotic arm.

**Figure 11 micromachines-15-00918-f011:**
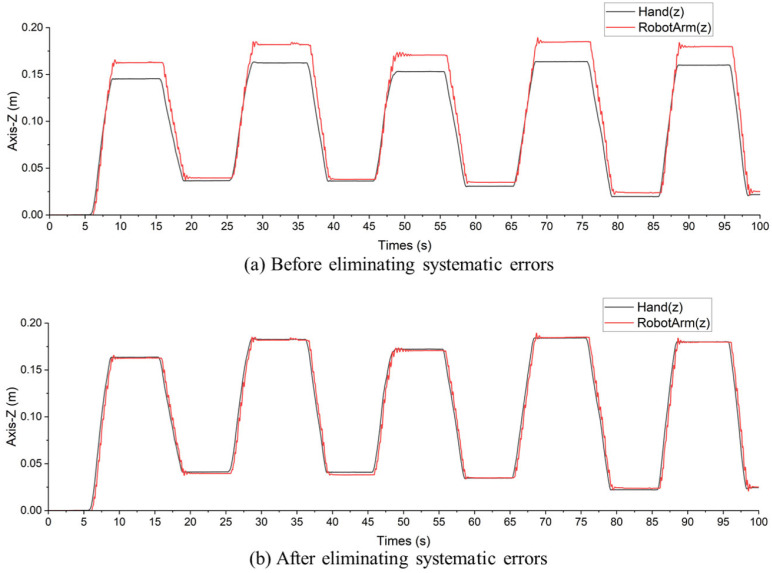
Data comparison before and after eliminating systematic errors. (**a**) Discrepancies between the robotic arm and human hand coordinates. (**b**) Corrected data demonstrating improved alignment after addressing systematic errors.

**Figure 12 micromachines-15-00918-f012:**
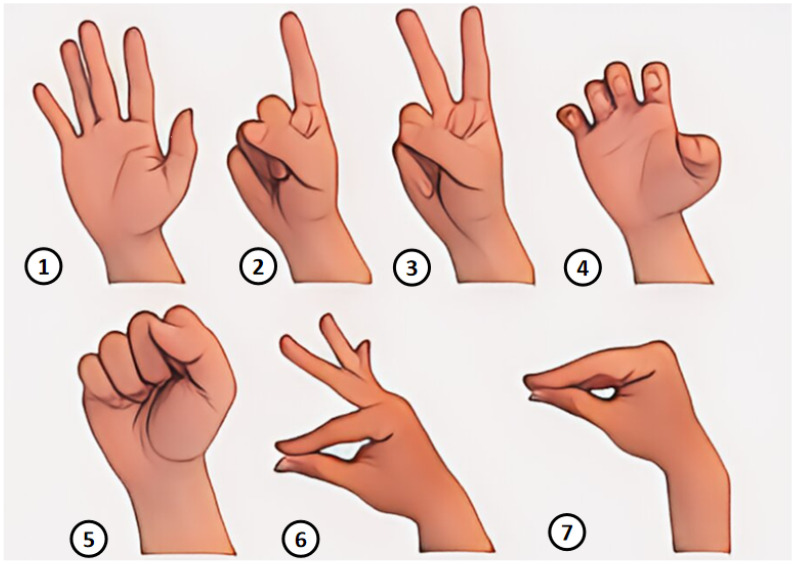
Gestures: ① rest, ② show 1, ③ show 2, ④ claw, ⑤ fist, ⑥ pinch with index finger and thumb, and ⑦ all-finger pinch.

**Figure 13 micromachines-15-00918-f013:**
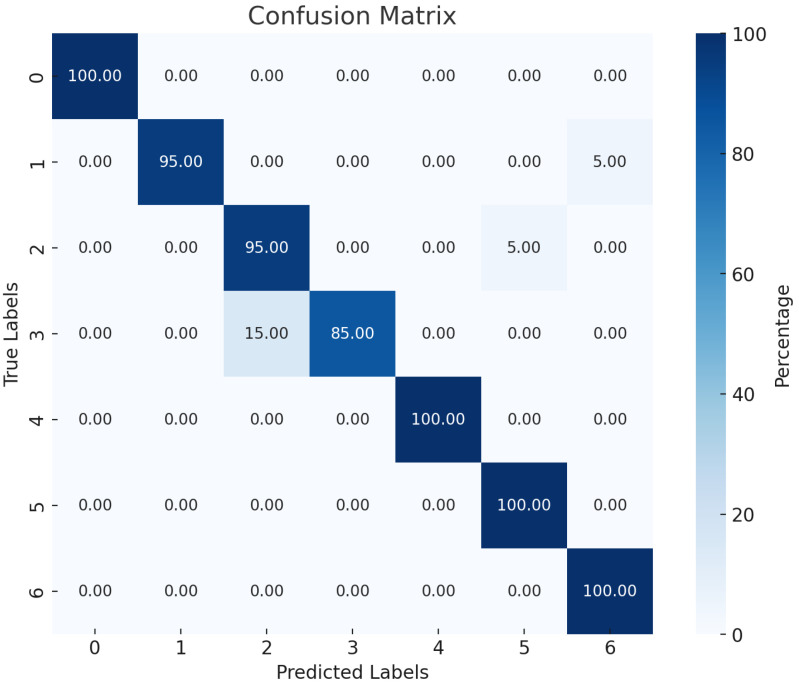
CNN+BiLSTM confusion matrix. (The darker the blue, the higher the recognition rate percentage).

**Figure 14 micromachines-15-00918-f014:**
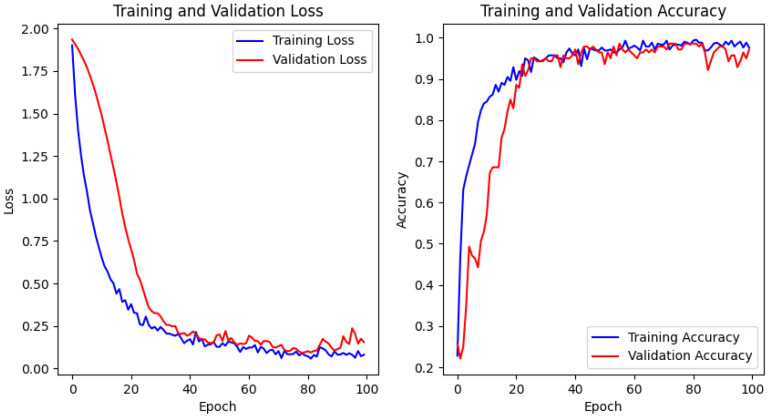
CNN+BiLSTM loss and accuracy.

**Table 1 micromachines-15-00918-t001:** Comparison of data gloves for hand gesture recognition.

Data Glove	Hand Gesture Sensor	Hand Position Sensor	Glove Costs	Reference
MIMU data glove	IMU sensor	IMU sensor	USD 200	[7]
Inertial sensor-based data glove	IMU sensor	IMU sensor	USD 500	[8]
HaptX Gloves G1 (HaptX Inc., Seattle, WA, USA)	IMU sensor	Magnetic capture system	USD 5495	[11]
XSENS PRIME 3 (MANUS Meta., Eindhoven, The Netherlands)	IMU and bending sensor	VR device camera	USD 4000	[12]
TactGlove DK2 (bHaptics, Daejeon, Republic of Korea)	VR device camera	VR device camera	USD 250	[13]
Bending sensor and OptiTrack-based data glove	Bending sensor	OptiTrack system	USD 100	This work

**Table 2 micromachines-15-00918-t002:** Performance metrics of different models for hand gesture recognition.

	Accuracy	Precision	Recall	F1-Score	Total Parameters
CNN+BiLSTM	0.9643	0.9669	0.9643	0.9642	365,511
BiLSTM	0.8857	0.8908	0.8857	0.8821	80,071
LSTM	0.6929	0.7229	0.6929	0.6878	19,655

## Data Availability

The data presented in this study are available from the corresponding author upon request.

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
