# Peer review of "Gesture-Controlled Robotic Arm for Agricultural Harvesting Using a Data Glove with Bending Sensor and OptiTrack Systems"

_micromachines, 2024, doi:10.3390/mi15070918_

Round 1

Reviewer 1 Report

Comments and Suggestions for Authors

This article introduces a cutting-edge robotic arm system that can be controlled through hand gestures, aiming to tackle the issues of labor shortage and physical demands in agricultural harvesting. The system includes a data glove with bending sensors and OptiTrack systems, and uses a CNN+BiLSTM model for accurate recognition of hand movements. The article showcases the system's innovative features and its potential for real-world use in both theoretical and practical settings. Some minor suggestions:

1. System Performance and Experimental Results:

o Supplement Experimental Data in Different Environments: It is recommended to conduct experiments under various environmental conditions (e.g., different lighting, temperature, and humidity) and with different types of crops (e.g., various shapes and sizes of fruits). Providing detailed experimental data will help verify the system's performance and stability in diverse real-world scenarios.

o Increase Long-term Experimental Data: It is suggested to perform long-term continuous operation experiments to assess the system's performance stability and durability over extended periods, particularly focusing on the sensors and robotic arm's performance during prolonged use.

2. System Limitations and Challenges:

o Discuss OptiTrack System Limitations in Outdoor Environments: Provide a detailed discussion on the potential issues the OptiTrack system might face in complex outdoor environments, such as occlusion problems and the effects of varying light conditions on reflective markers. Propose potential solutions, such as using multi-sensor fusion technology or improving the design of reflective markers.

o Integration of IMU with OptiTrack: Explore the possibility of integrating IMU (Inertial Measurement Unit) with the OptiTrack system to leverage IMU's dynamic measurement advantages and OptiTrack's high-precision positioning capabilities, thereby enhancing the system's performance in dynamic and complex environments.

Reviewer 2 Report

Comments and Suggestions for Authors

1. It is not clear what is the key message that authors want to convey in this paper. In other words, problem and solution messaging is not clear and convincing to reader.

What is the problem that you solve?

How big is the problem?

What are the existing solutions?

How bad the existing solutions are?

2. Haptic glove is very common topics, A literature comparison with previous efforts using a comparison table should be displayed.

3. Haptic glove is commercially available, even with wireless transmission. What is the advantage and disadvantage of the developed system compare with commercially available system, for example, Haptic gloves from HaptX, VR gloves from Manus Meta, TactGlove DK2 glove from b Haptic, and the price as low as $250. The commercially available glove looks more robust and smaller than the developed system. How do you convince readers that your system is better than commercially available glove?

4. How reliable of the developed system? What is the repeatability of the developed system, please also provide the standard deviation.

5. Author has showing the bending effect of the commercial sensor that used for develop the system. But wear the glove and bend the finger, the sensor will also have tensile strength. What is the sensor behavior during the tensile strength/strain, this experiment is missing. Please following the paper below for the experiment.

A. K. Bose et al., "Screen-Printed Strain Gauge for Micro-Strain Detection Applications," in IEEE Sensors Journal, vol. 20, no. 21, pp. 12652-12660

6. Please use third person and passive-voice constructions rather than First-Person Pronouns.

In general, this work seems to be interesting and can be considered for acceptance after address all the comments. I dope hope to see the revised review.

Comments on the Quality of English Language

English is acceptable
